# Physical Inactivity, Metabolic Syndrome and Prostate Cancer Diagnosis: Development of a Predicting Nomogram

**DOI:** 10.3390/metabo13010111

**Published:** 2023-01-09

**Authors:** Cosimo De Nunzio, Aldo Brassetti, Fabiana Cancrini, Francesco Prata, Luca Cindolo, Petros Sountoulides, Chrysovalantis Toutziaris, Mauro Gacci, Riccardo Lombardo, Antonio Cicione, Giorgia Tema, Luigi Schips, Giuseppe Simone, Sergio Serni, Andrea Tubaro

**Affiliations:** 1Department of Urology, Sant’Andrea Hospital, “Sapienza” University of Rome, Via di Grottarossa 1035, 00189 Rome, Italy; 2Department of Urology, IRCCS “Regina Elena” National Cancer Institute, Via Elio Chianesi 53, 00144 Rome, Italy; 3Department of Urology, “Padre Pio da Pietrelcina” Hospital, Via S. Camillo del Lellis, 66054 Vasto, Italy; 4Department of Urology, General Hospital of Veria, Synoikismos Papagou, 59132 Veria, Greece; 5Department of Urology, Aristotle University of Thessaloniki, 54124 Thessaloniki, Greece; 6Department of Urology, Careggi Hospital, University of Florence, Largo Brambila 3, 50134 Florence, Italy

**Keywords:** high-grade tumor, metabolic syndrome, nomogram, physical activity, prostate cancer

## Abstract

Insufficient physical activity (PA) may be a shared risk factor for the development of both metabolic syndrome (MetS) and prostate cancer (PCa). To investigate this correlation and to develop a nomogram able to predict tumor diagnosis. Between 2016 and 2018, a consecutive series of men who underwent prostate biopsy at three institutions were prospectively enrolled. PA was self-assessed by patients through the Physical Activity Scale for the Elderly (PASE) questionnaire; MetS was assessed according to Adult Treatment Panel III criteria. A logistic regression analyses was used to identify predictors of PCa diagnosis and high-grade disease (defined as International Society of Uro-Pathology grade >2 tumors). A nomogram was then computed to estimate the risk of tumor diagnosis. A total of 291 patients were enrolled; 17.5% of them (*n* = 51) presented with MetS. PCa was diagnosed in 110 (38%) patients overall while 51 presented high-grade disease. At multivariable analysis, age (OR 1.04; 95%CI: 1.00–1.08; *p* = 0.048), prostate volume (PV) (OR 0.98; 95%CI: 0.79–0.99; *p* = 0.004), suspicious digital rectal examination (OR 2.35; 95%CI: 1.11–4.98; *p* = 0.02), total PSA value (OR 1.12; 95%CI: 1.05–1.2; *p* < 0.001), and PASE score (OR 0.99; 95%CI: 0.98–0.99; *p* = 0.01) were independent predictors of tumor diagnosis. The latter two also predicted high-grade PCa. MetS was not associated with PCa diagnosis and aggressiveness. The novel nomogram displayed fair discrimination for PCa diagnosis (AUC = 0.76), adequate calibration (*p* > 0.05) and provided a net benefit in the range of probabilities between 20% and 90%. reduced PA was associated with an increased risk of PCa diagnosis and high-grade disease. Our nomogram could improve the selection of patients scheduled for prostate biopsy at increased risk of PCa.

## 1. Introduction

Evidence supporting the benefits of physical activity (PA) has grown and a sedentary lifestyle is now considered a public health plague in Western countries [1]. Epidemiological projections are not reassuring as it has been estimated that the incidence of obesity, hypertension and diabetes will dramatically increase in the next decade [2]. These diseases of affluence are key components of metabolic syndrome (MetS), a pro-inflammatory systemic condition which not only increases by twofold the risk of cardiovascular disorders [3], but is also associated with an increased incidence of different cancers in men and women [4].

Prostate cancer (PCa) is the most common malignancy in men, with 268,490 estimated new cases in 2022 in the U.S. [5]. Although several studies have evaluated the possible role of genetic/dietary factors and PA in PCa development and progression, to date, no specific preventive or dietary measure is recommended to reduce the risk of developing this neoplasm [6]. A high caloric diet combined with a sedentary lifestyle could lead to obesity, diabetes, dyslipidemia, hypertension and even MetS, which in turn may be associated with a higher risk of PCa diagnosis and aggressiveness [3]. 

As the role of PA and MetS in PCa development and progression is still debated and evidence is based on cohort studies with conflicting results [7,8,9,10], we aimed at investigating the association between an active lifestyle (assessed by means of the internationally validated Physical Activity Scale for the Elderly [PASE]), *Syndrome X* and PCa diagnosis and aggressiveness in a consecutive multicenter series of men undergoing prostate biopsy. We also developed a clinical nomogram to predict the risk of being diagnosed with this malignancy.

## 2. Materials and Methods

### 2.1. Study Population and Design

The study protocol was approved by the institutional committee on human research, ensuring that it conformed to the ethical guidelines of the 1975 Declaration of Helsinki. Written informed consent was obtained from all patients. From September 2016 to September 2018, all the men with at least a 10–15-year life expectancy and a clinical suspicion of PCa (because of a prostate specific antigen [PSA] value ≥ 4 ng/mL and/or an abnormal digital rectal examination [DRE]) referred to the urology departments of the five participating institutions were scheduled for biopsy and prospectively enrolled in the study, by signing a dedicated informed consent. Patients taking 5-phosphodiesterase inhibitors (5ARi) and those with a personal history of benign prostatic hyperplasia (BPH) surgery were excluded. A conventional 12-core transrectal ultrasound (TRUS)-guided biopsy [11] was performed: a 5–10 MHz bi-convex probe was used, together with a 16-gauge biopsy needle and a spring-loaded biopsy gun. Antibiotic prophylaxis and periprostatic anesthetic block were delivered according to internal protocols [12]. Prostate volume (PV) was ultrasonographically assessed at the time of biopsy, resorting to the ellipsoid formula [13]. One experienced uro-pathologist in each center, blinded for all clinical data except age, assessed the specimens and assigned the Gleason Score: International Society of Uro-Pathology (ISUP) grade > 2 tumors were defined as high-grade PCa (HG-PCa) [14]. 

### 2.2. Assessments of Metabolic Status and Physical Activity

Age and anthropometric parameters such as height, weight, and waist circumference (measured using a standard measurement strip, midway between the lowest rib margin and the iliac crests) were assessed in all patients [1,2]. Body mass index (BMI) was calculated as weight in kilograms divided by height in meters, squared: a conventional cut-off of 30 kg/m^2^ was used to define obesity. Resting blood pressure was recorded as the first and fifth Korotkof sounds by auscultation methods [15]. To ensure the accuracy of the estimation, measurements were performed three consecutive times by trained staff members (resident doctors, nurses); a discordance rate <5% was considered acceptable. Before urethral or prostate manipulation, fasting blood samples were collected and tested for glucose, cholesterol, triglycerides and total PSA levels. Serum concentration of the kallikrein-3 was determined using the Roche Elecsys^®^ total PSA assay (Hoffmann-La Roche, Basel, Switzerland), whose blank and detection limits are 0.006 ng/mL and 0.014 ng/mL, respectively, with an estimated sensitivity of 92.4% and a negative predictive value of 65.4% [16]. MetS was eventually diagnosed according to the Adult Treatment Pannell III (ATPIII) criteria [17]. The average PA was self-assessed by the patients through the PASE score [18], a validated 11-item questionnaire that combines information on leisure, household and occupational activities, which was adopted in other similar studies investigating the possible association between exercise and prostatic diseases [3,19,20]. The obtained scores were further stratified into 3 classes to define low (<100), moderate (101–250) and high (>250) levels of PA [3].

### 2.3. Statistical Analysis

The study population was split into two groups according to PCa diagnosis/HG-PCa. Frequencies and proportions were used to report categorical variables that were compared by means of the Chi-squared test. Continuous variables were presented as median and interquartile ranges (IQRs) and were compared using the Kruskal–Wallis test. Logistic regression models were used to identify predictors of PCa diagnosis and high-grade disease. Coefficients were then used to generate a nomogram to predict tumor diagnosis, whose calibration was assessed using the Hosmer–Lemeshow test. Calibration plots were generated with 200 bootstraps resampling to explore nomogram performance, and decision curve analyses (DCAs) assessed the net benefit of the model. For all tests, the significance level was set at a *p* value of <0.05. Statistical analysis was performed using the Statistical Package for Social Science v. 24.0 (IBM, Somers, NY, USA), the R statistical package (The R-Project for Statistical Computing, www.r-project.org) and STATA (StataCorp. 2019. Stata Statistical Software: Release 16. College Station, TX, USA: StataCorp LLC).

## 3. Results

### 3.1. Patient Characteristics

Overall, 291 patients were enrolled (with a median age of 65 years [IQR 60/71]) and 51 (17%) presented with MetS. The median PASE score was 100 (IQR: 93/109) and 43% of the included men reported a sedentary lifestyle. A tumor was diagnosed in 110 (38%) patients overall (Table 1). 

PCa men were older (70 years vs. 65 years), more sedentary (median PASE: 95 vs. 128) and presented with smaller prostates (43 mL vs. 50 mL) and higher PSA values (8 ng/mL vs. 6.71 ng/mL) (all *p* < 0.001). The rate of MetS was comparable between PCa and non-PCa patients (19% vs. 16%; *p* = 0.35). Logistic regression analyses identified age (OR 1.04; 95%CI: 1.00–1.08; *p* = 0.048), total PSA value (OR 1.12; 95%CI: 1.05–1.2; *p* < 0.001), PV (OR 0.98; 95%CI: 0.79–0.99; *p* = 0.004), suspicious digital rectal examination (OR 2.35; 95%CI: 1.11–4.98; *p* = 0.02) and PASE score (OR 0.99; 95%CI: 0.98–0.99; *p* = 0.01) as independent predictors of tumor diagnosis (Table 2). Out of the 110 identified tumors, 51 (46%) were high-grade. Patients with an aggressive disease showed higher total PSA levels (9 ng/mL vs. 7 ng/mL; *p* = 0.018) and lower PASE scores (93 vs. 104; *p* = 0.01) (Table 1) compared to those harboring a low-grade tumor. No difference in terms of MetS prevalence was observed in the two groups. At logistic regression analyses, total PSA values (OR 1.05; 95%CI: 1.00–1.10; *p* = 0.047) and PASE scores (OR 0.99; 95%CI: 0.98–0.99; *p* = 0.02) were independent predictors of HG-PCa (Table 2).

### 3.2. Model Characteristics

The novel nomogram presented fair discrimination for PCa diagnosis (AUC = 0.76), adequate calibration (*p* > 0.05) and provided a net benefit in the range of probabilities between 20% and 90% (Figure 1). Internal validation after 200 bootstraps showed an AUC of 0.75.

## 4. Discussion

In Western countries, MetS and PCa are among the most compelling health challenges, and their incidence is deemed to increase [2] because of population ageing and sedentary lifestyle [21]. These diseases may share etiological agents: in fact, the former is characterized by a systemic inflammatory status, and the latter was recently associated with elevated circulating inflammatory markers [22,23,24,25]. Diabetes mellitus, hypertension and obesity are responsible for a chronic low-grade *metaflammation* [26], which in turn may induce tumorigenesis through direct genomic damage, local immunosuppression and promotion of cell proliferation [27,28]. At least one MetS component is found in most men with elevated PSA levels [29]. Patients with hypertension are at increased risk of developing PCa [30], and calcium channel blockers are thought to promote carcinogenesis by affecting the Cav 3.1 channels, which are involved in tumor suppression and apoptosis promotion [31]. Interestingly, although advanced glycation end products may cause cellular dysfunction [26], diabetes seems inversely associated with the risk of PCa [32], though this observation can be explained by cancer underdiagnosis due to low PSA levels that characterize most men with type 2 diabetes [33,34]. Conflicting results were reported concerning the impact of obesity on PCa risk. In fact, it was hypothesized that an imbalance between serum concentration of estrogens, testosterone, insulin and insulin-like growth factor-1, which is distinctive of obese patients, could promote carcinogenesis [35]. However, recent reviews failed to detect an association between visceral obesity and PCa development [36]. Again, these observations might be explained by underdiagnosis, as BMI is inversely associated with PSA levels [37]. 

Although conflicting results were reported, a recent meta-analysis (which included data of 132,589 patients) concluded that MetS is associated with an increased risk of HG-PCa at biopsy, adverse features at final pathology, disease recurrence and cancer-specific mortality [9,10,23,38,39]. Conversely, results from the present study failed to prove any association between MetS and PCa diagnosis and high-grade disease (Table 2). 

Grounded evidence support the protective role of PA on the risk of MetS, while its possible association with PCa prevention has been recently investigated and conflicting results were reported [40,41]. It is known that exercise promotes telomere enlengthening, modulates genes expression and affects proteins intracellular transportation, metabolism and phosphorylation [42]. Improving insulin resistance, PA interferes with the serum levels of several tumor-promoting proteins such as insulin-like growth factor-1 [43] and, while reducing adiposity, it decreases blood levels of proinflammatory adipokines [44]. An active lifestyle reduces the risk of PCa diagnosis and high-grade tumor at biopsy [3,11] and even reclassification during active surveillance [8]. Increasing levels of PA are associated with a reduced risk for tumor progression, recurrence and disease-specific death after primary treatment [45,46]. For these reasons, exercise has been included in the ASCO (American Society of Clinical Oncology) Clinical Practice Guidelines on PCa [47]. Results from the present study confirmed the previously mentioned findings, as each increase in PASE was associated with a significant reduction in the risk of PCa diagnosis and aggressive disease (Table 2).

After an abnormal screening test, many men are advised to undergo prostate biopsy and wish to know the odds of harboring PCa. Physicians may need to estimate the probability of a positive biopsy before recommending it. However, referring urologists may have difficulty in assessing the risk for individual patients: in fact, while several risk factors for PCa have been identified, their combined contribution may be difficult to estimate. Nomograms have been conceived to quantify the pooled contribution of various risk factors and provide a predicted probability of the event of interest which applies to an individual instead of situating this individual within a generic risk group. Studies have clearly shown that these models predict more accurately than expert clinicians; thus, it is likely that the advantage of using them may be even more significant if clinical ratings are obtained from average doctors. To reduce the number of men requiring a prostate biopsy, the European Association of Urology encourages physicians to resort to one of the nomograms [48] developed in the past few decades (with an AUC of 0.69–0.75), as none have proven superiority compared to the others and head-to-head comparisons are virtually lacking [49]. Most of them, however, were developed in the sextant biopsy era and may not be able to predict the probability of PCa on needle biopsy in the present extended biopsy era [50]. Our model showed fair accuracy (AUC = 0.76) and sufficient DCA. Interestingly, ours is the first predicting nomogram that includes PA among the other variables, considering its previously discussed association with tumor diagnosis and high-grade disease. The main limitation of our model is the lack of magnetic resonance imaging (MR) data. According to the most recent guidelines, in fact, performing an MRI scan before prostate biopsy is strongly recommended as it results in avoiding 30% of all procedures while missing 11% of ISUP grade >2 cancers [51]. However, at the time of the present study, MR was not mandatory yet. Moreover, it must be considered that systematic biopsy is still an acceptable approach in case MRI is unavailable [48].

We must also acknowledge limitations related to our study design. First, our results obviously depend upon the enrolled population and the observed average PA, which may differ from other countries. In this regard, however, one should consider that ours is a multicenter international study—the participation of men from different European countries at least mitigates the inherent selection bias. Another possible limitation derives from the use of a biopsy cohort without confirmation from radical prostatectomy specimens: in fact, up to one third of PCa is found upgraded at final pathology. Concerning this, though, there is evidence that the rate of upgrading increases with PSA density, which was extremely low in the present study (0.06 ng/mL^2^) [51]. Another possible limitation of the present study is the use of the Physical Activity Scale for the Elderly to measure the average daily energy expenditure: this tool, in fact, was initially conceived for patients older than 65 years, and a quarter of our study population was younger than 60 years. This questionnaire, however, is not only indicated for retired men, as it investigates occupational, household and leisure activities. In fact, it was already successfully administered to cohorts of patients of the same age [8,11,52,53] or even younger [54,55]. One last limitation of the present study is that sex hormone levels and testosterone replacement therapies (TRT) were not assessed at the time of prostate biopsy; consequently, their association with PCa and MetS was not investigated. However, a strong epidemiologic association between hypogonadism, MetS and its features was recently observed [56]. Besides, there is grounded evidence that sex hormones play a key role in the development and progression of PCa, although the pathogenetic mechanisms was not clarified yet [57]. Moreover, the influence of TRT on tumorigenesis is still debated [58], with recent evidence supporting that it has little, if any, negative impact on the prostate, even in men with a history of PCa [57,59,60].

Notwithstanding these limitations, ours is the largest multicenter prospective series in Europe evaluating the association between PA, MetS and PCa risk, using the same biopsy template for the entire cohort. We also conceived the first nomogram to estimate the risk of PCa diagnosis that includes PA among the other conventional predicting variables. 

## 5. Conclusions

A sedentary lifestyle is extremely common in Western countries. According to our cohort study on Southern European men, this behavior is associated with a significantly increased risk of PCa diagnosis and high-grade disease. However, molecular mechanisms behind this association deserve a more in-depth investigation. Besides, no association was found between MetS and PCa incidence and aggressiveness.

If externally validated, our nomogram may represent a reliable and easy tool to assess PCa risk in daily practice (particularly if MRI is not available) which considers not only conventional clinical variables but also patients’ physical activity.

## Figures and Tables

**Figure 1 metabolites-13-00111-f001:**
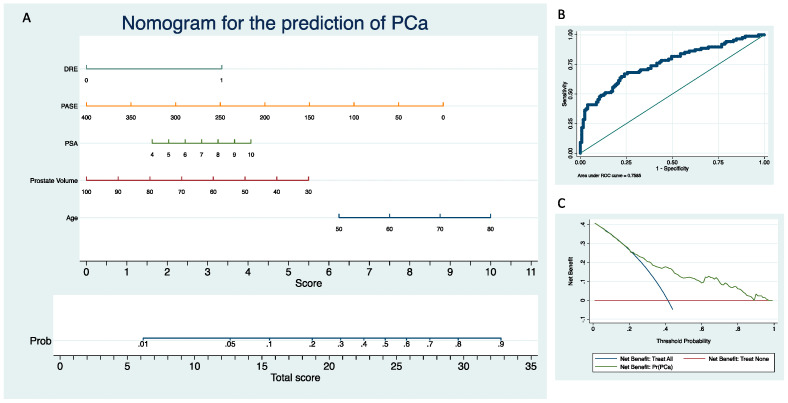
Clinical nomogram to predict prostate cancer diagnosis (**A**) and its related ROC curve (**B**) and DCA (**C**).

**Table 1 metabolites-13-00111-t001:** Patients characteristics according to PCa diagnosis and high-grade disease.

	Overall (n = 291)	No PCa (n = 181)	PCa (n = 110)	*p*	LG-PCa (n = 59)	HG-PCa (n = 51)	*p*
Age, years	65 (60/71)	65 (60/71)	70 (63/75)	**<0.001**	70 (62/65)	71 (65/76)	0.36
BMI	26.9 (24/29.2)	26.5 (24.5/29)	26.9 (25/29.2)	0.64	26 (25/29)	28 (25/29)	0.34
MetS, n (%)	51 (17%)	30 (16%)	21 (19%)	0.35	12 (20%)	9 (18%)	0.72
PSA, ng/mL	6.7 (4.7/9)	6.71 (4.8/8.7)	8 (5.7/13.6)	**<0.001**	7 (5/10)	9 (6/15)	**0.02**
PV, mL	48 (35/68)	50 (39/68)	43 (28/55)	**<0.001**	42 (28/53)	43 (27/57)	0.73
PASE score	100 (93/109)	128 (86/194)	95 (58.7/152)	**<0.001**	104 (68/170)	93 (35/127)	**0.01**

Data are presented as median (IQR). Statistically significant differences are shown in bold font. PCa = prostate cancer, LG-PCa = low-grade prostate cancer, HG-PCa = high-grade prostate cancer, BMI = body mass index, PSA = prostate-specific antigen, MetS = metabolic syndrome, PASE = Physical Activity Scale for the Elderly, PV = prostate volume.

**Table 2 metabolites-13-00111-t002:** Logistic regression analyses to identify predictors of prostate cancer diagnosis and high-grade disease.

	PCa Diagnosis								HG-PCa							
	Univariable Analysis				Multivariable Analysis				Univariable Analysis				Multivariable Analysis			
	OR	95% CI		*p*	OR	95% CI		*p*	OR	95% CI		*p*	OR	95% CI		*p*
Age	**1.07**	**1.03**	**1.10**	**<0.001**	**1.04**	**1.00**	**1.08**	**0.048**	1.02	0.97	1.07	0.33	-	-	-	-
BMI	1.02	0.96	1.09	0.51	-	-	-	-	1.05	0.96	1.15	0.32	-	-	-	-
MetS	1.19	0.64	2.20	0.58	-	-	-	-	0.84	0.32	2.20	0.72	-	-	-	-
PSA	**1.12**	**1.06**	**1.17**	**<0.001**	**1.12**	**1.05**	**1.2**	**<0.001**	**1.05**	**1.01**	**1.11**	**0.04**	**1.05**	**1.00**	**1.10**	**0.047**
Prostate Volume	**0.98**	**0.97**	**0.99**	**0.01**	**0.98**	**0.79**	**0.99**	**0.004**	1.01	0.98	1.02	0.60	-	-	-	-
Suspicious DRE	**3.07**	**1.69**	**5.57**	**<0.001**	**2.35**	**1.11**	**4.98**	**0.02**	1.74	0.77	3.93	0.18	-	-	-	-
PASE score	**0.99**	**0.98**	**0.99**	**<0.001**	**0.99**	**0.98**	**0.99**	**0.01**	**0.99**	**0.98**	**0.99**	**0.01**	**0.99**	**0.98**	**0.99**	**0.02**

Statistically significant differences are shown in bold font. PCa = prostate cancer, HG-PCa = high-grade prostate cancer, BMI = body mass index, MetS = metabolic syndrome, PSA = prostate-specific antigen, PASE = Physical Activity Scale for the Elderly, DRE = digital rectal examination.

## Data Availability

The metabolomics and metadata reported in this paper are available from the corresponding author upon request. Data is not publicly available due to privacy or ethical restrictions.

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
