# Peer review of "Physical Inactivity, Metabolic Syndrome and Prostate Cancer Diagnosis: Development of a Predicting Nomogram"

_metabolites, 2023, doi:10.3390/metabo13010111_

Round 1
Reviewer 1 Report
Dear Authors,
I have reviewed the article titled, Physical inactivity, Metabolic Syndrome and Prostate Cancer diagnosis: development of a predicting nomogram. Which is a study that aims to relate physical activity and metabolic syndrome with prostate cancer.
Below I list my observations and comments.
1.1 The objectives should be stated in the infinitive, writing in the third person should be sought (Lines 17-18).
1.2 Nothing is indicated about the evaluation of the metabolic syndrome.
1.3 Instead of putting the frequency, the prevalence of the metabolic syndrome can be indicated (line 24).
1.4 in line 26 does not define what PV is.
1.5 The results are confusing since it is not indicated which marker is being contrasted.
1.1 The objectives should be stated in the infinitive, writing in the third person should be sought (Lines 17-18).
1.2 Nothing is indicated about the evaluation of the metabolic syndrome.
1.3 Instead of putting the frequency, the prevalence of the metabolic syndrome can be indicated (line 24).
1.4 in line 26 does not define what PV is.
1.5 The results are confusing since it is not indicated which marker is being contrasted.
1.6 In the conclusion, determining words must be used and if something is assumed it should be put into perspective. avoid using words like “seems” “might”
1.7 On the line indicate the abbreviation previously exposed for metabolic syndrome.
1.8 Lines 59-60 could be integrated into the objective of the work.
1.9 Describe the demographic characteristics of the population.
1.10 Declare whether the Helsinki protocol guidelines were followed
1.11 Describe the methodology used to determine prostate antigen, its sensitivity and specificity.
1.12 There is no mention of blood pressure measurement.
1.13 Describe the anthropometric methods. Were they taken by trained personnel?; How many times was the measurement taken?; What was the accepted error rate?
1.14 Describe what the PASE questionnaire consists of and what aspects it evaluates.
1.15 Indicate which were the independent and dependent variables.
1.16 If the population was divided according to metabolic syndrome (line 90), why do tables 1 and 2 show the classification according to the presence of prostate cancer and its severity?
1.17 Explain the clinical relevance of the Nomogram; describe the results clearly.
1.18 Authors must compare their results with the information cited.
1.19 The conclusion does not refer to the results, since the origin of the population studied is not mentioned.
Author Response
- The objectives should be stated in the infinitive, writing in the third person should be sought (Lines 17-18).
We thank the reviewer for the suggestion.
The abstract section concerning the objectives of the study has been modified as suggested:
- (Abstract, pg 1, lines 14-17)
Introduction: Insufficient Physical activity (PA) may be a shared risk factor for the development of both metabolic syndrome (MetS) and prostate cancer (PCa).
Objectives: To investigate this correlation and to develop a nomogram able to predict tumor diagnosis.
- Nothing is indicated about the evaluation of the metabolic syndrome.
We thank the reviewer for his/her comment.
A concise statement concerning how MetS was assessed has been added in the abstract:
- (Abstract, pg 1, lines 18-21)
Methods: Between 2016 and 2018, a consecutive series of men who underwent prostate biopsy at 3 institutions were prospectively enrolled. PA was self-assessed by patients through the physical activity scale for the elderly (PASE) questionnaire; MetS was assessed according to Adult Treatment Panel III criteria.
- Instead of putting the frequency, the prevalence of the metabolic syndrome can be indicated (line 24).
We thank the reviewer for his/her suggestion.
According to him/her and to Reviewer #2 requests, the text was amended as follows:
- (Abstract, pg 1, line 23)
Results: 291 patients were enrolled; 17.5% of them (n=51) presented with MetS.
- in line 26 does not define what PV is.
We thank the reviewer for the suggestion. PV defitition was spelled out at its first mention:
- (Abstract, pg 1, lines 24-28)
At multivariable analysis, age (OR 1.04; 95%CI: 1.00-1.08; p=0.048), prostate volume (PV) (OR 0.98; 95%CI: 0.79-0.99; p=0.004), suspicious digital rectal examination (OR 2.35; 95%CI: 1.11-4.98; p=0.02), total PSA value (OR 1.12; 95%CI: 1.05-1.2; p<0.001) and PASE score (OR 0.99; 95%CI: 0.98-0.99; p=0.01) were independent predictors of tumor diagnosis.
- The results are confusing since it is not indicated which marker is being contrasted.
We thank the reviewer for the comment.
According to his/her previous suggestions, we modified the abstract clarifying that the objective of our study was to investigate the correlation between PA, MetS and PCa and to develop a nomogram able to predict tumor diagnosis. Accordingly, we better elucidated in the Results section that, based on our analysis, MetS was not associated with PCa diagnosis and aggressiveness while PA independently predicted both. We also reported that our novel nomogram, which includes PA assessment through the PASE score, displayed fair discrimination for PCa diagnosis (AUC= 0.76).
- (Abstract, pg 1, lines 24-31)
At multivariable analysis, age (OR 1.04; 95%CI: 1.00-1.08; p=0.048), prostate volume (PV) (OR 0.98; 95%CI: 0.79-0.99; p=0.004), suspicious digital rectal examination (OR 2.35; 95%CI: 1.11-4.98; p=0.02), total PSA value (OR 1.12; 95%CI: 1.05-1.2; p<0.001), and PASE score (OR 0.99; 95%CI: 0.98-0.99; p=0.01) were independent predictors of tumor diagnosis. The latter two also predicted high-grade PCa. MetS was not associated with PCa diagnosis and aggressiveness. The novel nomogram displayed fair discrimination for PCa diagnosis (AUC= 0.76), good calibration (p>0.05) and provided a net benefit in the range of probabilities between 20% and 90%.
- In the conclusion, determining words must be used and if something is assumed it should be put into perspective. avoid using words like “seems” “might”
We thank the reviewer for his/her suggestion. The manuscript has been improved accordingly:
- (Abstract, pg 1, lines 32-34)
Conclusion: A reduced PA resulted to be associated with an increased risk of PCa diagnosis and high-grade disease. Our nomogram could improve the selection of patients scheduled for prostate biopsy at increased risk of PCa.
- On the line indicate the abbreviation previously exposed for metabolic syndrome.
We thank the reviewer for the suggestion.
Although the same abbreviation was already used and spelled out in the Abstract (line 16), we preferred spelling it out again in the manuscript, in the Introduction section, at its first mention:
- (Introduction, pg 1-2, lines 46-50)
These diseases of affluence are key-components of metabolic syndrome (MetS), a pro-inflammatory systemic condition which not only increases by two folds the risk of cardiovascular disorders [3] but was also associated with an increased incidence of different cancers in both men and women [4].
- Lines 59-60 could be integrated into the objective of the work.
We thank the reviewer for his/her suggestion. The introduction section has been modified as follows:
- (Introduction, pg 2, lines 59-64)
As the role of PA and MetS in PCa development and progression is still debated and evidences are based on cohort studies with conflicting results [7–10], we aimed at investigating the association between an active lifestyle (assessed by means of the internationally validated Physical Activity Scale for the Elderly [PASE]), Syndrome X and PCa diagnosis and aggressivenes in a consecutive multicenter series of men undergoing prostate biopsy. We also developed a clinical nomogram to predict the risk of being diagnosed with this malignancy.
- Describe the demographic characteristics of the population.
We thank the reviewer for the comment.
As suggested, we clarified that “all the men with at least 10-15 years life expectancy and a clinical suspicion of PCa […] were […] enrolled in the study”. (Materials and methods – study population and design, pg 2, lines 66-72).
Moreover, patients’ age is now clearly reported also in the Results section (pg 3, lines 119-120): “Overall, 291 patients were enrolled (with a median age of 65 years [IQR 60/71]) and 51 (17%) presented with MetS.”
Furthermore, median age was already displayed in Table 1.
- Declare whether the Helsinki protocol guidelines were followed
We thank the reviewer for the comment.
According to his/her suggestion, the manuscript was amended as follows:
- (Materials and methods, pg 2, lines 69-71)
The study protocol was approved by the institutional committee on human research, ensuring that it conformed to the ethical guidelines of the 1975 Declaration of Helsinki. Written informed consent was obtained from all patients.
- (Institutional Ethics Committee Statement, pg 4, lines 254-256)
The study protocol was approved by the institutional committee on human research, ensuring that it conformed to the ethical guidelines of the 1975 Declaration of Helsinki.
- Describe the methodology used to determine prostate antigen, its sensitivity and specificity.
We thank the reviewer for the suggestion.
Accordingly, the manuscript was modified as follows:
- (Materials and methods, pg 3, lines 97-102)
Before urethral or prostate manipulation, fasting blood samples were collected and tested for glucose, cholesterol, triglycerides and total PSA levels. Serum concentration of the kallikrein-3 was determined using the Roche Elecsys® total PSA assay (Hoffmann-La Roche, Basel, Switzerland), whose Blanck and Detection limits are 0.006 ng/ml and 0.014 ng/ml respectively, with an estimated sensitivity of 92.4% and negative predictive value of 65.4%[13]
1.12 There is no mention of blood pressure measurement
1.13 Describe the anthropometric methods. Were they taken by trained personnel?; How many times was the measurement taken?; What was the accepted error rate?
We thank the reviewer for these comments.
A detailed description of our measurement methods was added to the manuscript.
- (Materials and methods, pg 2-3, lines 89-97)
Age and anthropometric parameters such as height, weight, and waist circumference (measured using a standard measurement strip, midway between the lowest rib margin and the iliac crests) were assessed in all patients [1,2]. Body mass index (BMI) was calculated as weight in kilograms divided by height in meters, squared: a conventional cut-off of 30 kg/m2 was used to define obesity [11]. Resting blood pressure was recorded as the first and fifth Korotkof sounds by auscultation methods[12]. To ensure the accuracy of the estimation, measurements were performed three consecutive times by trained staff members (resident doctors, nurses); a discordance rate < 5% was considered acceptable.
1.14 Describe what the PASE questionnaire consists of and what aspects it evaluates.
We thank the reviewer for the suggestion. The manuscript was modified as follows:
- (Materials and methods, pg 3, lines 103-117)
The average PA was self-assessed by the patients through the PASE score [18], a validated 11-ithems questionnaire that combines information on leisure, household, and occupational activities, which was already adopted in other similar studies investigating the possible association between exercise and prostatic diseases [3,19,20]
1.15 Indicate which were the independent and dependent variables.
1.16 If the population was divided according to metabolic syndrome (line 90), why do tables 1 and 2 show the classification according to the presence of prostate cancer and its severity?
We thank the reviewer for these comments.
We totally agree with him/her that the aim of the present study required clarification as well as the description of the statistical analyses that we performed.
Therefore, the dedicated sections were modified as follow:
- (Introduction, pg 2, lines 59-64)
As the role of PA and MetS in PCa development and progression is still debated and evidences are based on cohort studies with conflicting results [7–10], we aimed at investigating the association between an active lifestyle (assessed by means of the internationally validated Physical Activity Scale for the Elderly [PASE]), Syndrome X and PCa diagnosis and aggressivenes in a consecutive multicenter series of men undergoing prostate biopsy. We also developed a clinical nomogram to predict the risk of being diagnosed with this malignancy.
- (Materials and methods – statistical analysis, pg 3, lines 111-)
The study population was split into two groups according to PCa diagnosis/HG-PCa. Frequencies and proportions were used to report categorical variables, that were compared by means of the Chi-squared test. Continuous variables were presented as median and interquartile ranges (IQRs) and were compared using the Kruskal–Wallis test. Logistic regression models were used to identify predictors of PCa diagnosis and high-grade disease.
As above clarified, PCa diagnosis and aggressiveness were the two dependent variables (effects) while the available baseline characteristics such as age, BMI, MetS, PSA, PV and PASE score were the independent ones (possible causes).
1.17 Explain the clinical relevance of the Nomogram; describe the results clearly.
We thank the reviewer for the comment. As suggested, we clarified the role of nomograms in PCa diagnosis, highlighted the advantages provided by the already available models in selecting patients that may benefit the most from a biopsy and emphasized advantages and disadvantages of the novel nomogram we conceived.
- (Discussion, pg 6, lines 197-222)
After an abnormal screening tests, many men are suggested to undergo prostate biopsy and wish to know the odds of harboring PCa. Also physicians may require estimating the probability of positive biopsy before recommending it. However, referring urologists may have difficulty in assessing the risk for individual patients: in fact, while several risk factors for PCa have been identified, their combined contribution may be difficult to estimate. Nomograms have been conceived to quantify the pooled contribution of various risk factors and provide a predicted probability of the event of interest which applies to an individual instead of situating this individual within a generic risk group. Studies have clearly shown that these models predict more accurately than expert clinicians, thus it is likely that the advantage of using them may be even more significant if clinical ratings are obtained from average doctors. To reduce the number of men requiring a prostate biopsy, the European Association of Urology encourages physicians to resort to one of the nomograms [48] developed in the last decades (with an AUC of 0.69-0.75), as none has proven superiority compared to the others and head‐to‐head comparisons are virtually lacking [49]. Most of them, however, were developed in the sextant biopsy era and may not be able to predict the probability of PCa on needle biopsy, in the present extended biopsy era [50]. Our model showed a fair accuracy (AUC = 0.76) and an excellent DCA. Interestingly, it is the first predicting nomogram that comprises PA among the other variables, considering its previously discussed association with tumor diagnosis and high-grade disease. The main limitation of our model is the lack of magnetic resonance imaging (MR) data. According to the most recent guidelines, in fact, performing an MRI scan before prostate biopsy is strongly recommended as it results in avoiding 30% of all procedures while missing 11% of ISUP grade > 2 cancers [51]. However, at the time of the present study, MR was not mandatory yet. Moreover, it must be considered that systematic biopsy is still an acceptable approach in case MRI is unavailable[48].
1.18 The conclusion does not refer to the results, since the origin of the population studied is not mentioned.
We thank the reviewer for the comment. As suggested, the origin of the study population was clarified in the Conclusion section:
- (Conclusion, pg 7, lines 238-241)
A sedentary lifestyle is extremely common in Western Countries. According to our cohort study on Southern European men, this behavior was associated with a significantly increased risk of PCa diagnosis and high-grade disease. However, molecular mechanisms behind this association deserve a more in-depth investigation.

Reviewer 2 Report
The study "Physical inactivity, Metabolic Syndrome and Prostate Cancer diagnosis: development of a predicting nomogram" by Nunzio et al. is an interesting work to predict prostate cancer in men with high BMI and low physical activity. The methodology of this manuscript is quite impressive but the writing and data presentation needs a lot of improvements. Please see attached file for specific comments, in brief, the abstract, introduction tables, figures, and conclusion need improvements.

Author Response
We thank the reviewer for comments and suggestions that he/she noted down directly on the manuscript.
All the amendments that we adopted can be found below, highlighted in cyan.
- The percentage of patients presenting with MetS was specified, according to Rev #1 and Rev #2 suggestions.
- (Abstract, pg 1, line 24)
Results: 291 patients were enrolled; 17.5% of them (n=51) presented with MetS.
- The definition of HG-PCa was clarified in the Methods section of the Abstract
- (Abstract, pg 1, lines 21-22)
A logistic regression analyses was used to identify predictors of PCa diagnosis and high-grade disease (defined as International Society of Uro-Pathology grade > 2 tumors).
- As suggested by Rev #2, the first sentence of the Materials and Methods section was divided into two sentences. At the same time, as required by Rev #1, we also clarified that the study was approved by the institutional committee on human research and conducted in accordance with the declaration of Helsinki
- (Materials and methods, pg 2, lines 65-72)
The study protocol was approved by the institutional committee on human research, ensuring that it conformed to the ethical guidelines of the 1975 Declaration of Helsinki. Written informed consent was obtained from all patients. From September 2016 to September 2018, all the men with at least 10-15 years life expectancy and a clinical suspicion of PCa (because of a prostate specific antigen [PSA] value ≥ 4 ng/ml and/or an abnormal digital rectal examination [DRE]) referred to the urology departments of the five participating institutions were scheduled for biopsy and prospectively enrolled in the study, by signing a dedicated informed consent.
- Table 1 was modified so that the number of patients and the related percentages are displayed with a consistent style.
Also the overall value of BMI was checked and found wrong: the correct value was reported
- The word “wile” was deleted from line 159
- (Discussion, pg 5, lines 159-161)
These diseases may share etiological agents: in fact, while the former is characterized by a systemic inflammatory status, the latter was recently associated with elevated circulating inflammatory markers [22–25]
- According to the suggestions received by Rev # 1 and #2, a more comprehensive conclusion was added:
- (Conclusions, pg 7, lines 254-258)
A sedentary lifestyle is extremely common in Western Countries. According to our cohort study on Southern European men, this behavior was associated with a significantly increased risk of PCa diagnosis and high-grade disease. However, molecular mechanisms behind this association deserve a more in-depth investigation. Besides, no association was found between MetS and PCa incidence and aggressiveness.
If externally validated, our nomogram may represent a reliable and easy tool to assess PCa risk in daily practice (particularly if MRI is not available) which considers not only conventional clinical variables but also patients’ physical activity.

Reviewer 3 Report
Dear Authors,
The manuscript titled "Physical inactivity, Metabolic Syndrome and Prostate Cancer diagnosis: development of a predicting nomogram" is well written and logically organized.
However, I have some minor concerns about your data and suggestions in order to improve the quality of your paper.
1- The introduction section in the abstract is very short. Furthermore, the sentence is misleading: "Physical activity" should be replaced with "Physical inactivity".
2- Concerning table 1, the prostate volume (PV) of all patients, the median is 118 and in the two groups, No PCa and PCa was 50 and 43. Is that correct?
3- Since the paper correctly mention the imbalance of different hormones (such as oestrogen and testosterone), did the Authors have data concerning the role of levels of these hormones in men with metabolic syndrome and/or prostate cancer? Moreover, since many disorders are associated with metabolic sindrome (doi : 10.2164/jandrol.108.006015), such as hypertension, hyperglycaemia and hypogonadism, and in this latter case testosterone administration has been suggested as potential therapeutic drugs, what is the Authors point of view about the therapeutic use of testosterone? Indeed, in the scientific literature, the testosterone administration and prostate cancer risk is still debated (doi: 10.1158/1055-9965.EPI-04-0715; doi: 10.1016/j.ucl.2007.08.002; doi: 10.1093/aje/kwz138; doi: 10.1177/1756287215597633).
4- Latin words and abbreviations should be written in italics, such as "vs" at line 109, page 3.
5- The p value should be written as < 0.05 or <0.01.
Author Response
- The introduction section in the abstract is very short. Furthermore, the sentence is misleading: "Physical activity" should be replaced with "Physical inactivity".
We thank the reviewer for the comment and the suggestion.
Indeed, the Introduction section of the abstract is very short, for the sake of brevity. However, we believe that, together with the emended version of the Objectives, it provides a good overview of the background of our research and of the aim of our study.
As recommended, the misleading term “Physical activity” was modified in “Insufficent Phisical Activity”. By doing so, 1) we used the exact term endorsed by the World Health Organization (https://www.who.int/data/gho/indicator-metadata-registry/imr-details/3416), 2) the abbreviation “PA” could still be used in the following lines of the abstract.
- (Abstract, pg 1, lines 14-17)
Introduction: Insufficient Physical activity (PA) may be a shared risk factor for the development of both metabolic syndrome (MetS) and prostate cancer (PCa).
Objectives: To investigate this correlation and to develop a nomogram able to predict tumor diagnosis.
- Concerning table 1, the prostate volume (PV) of all patients, the median is 118 and in the two groups, No PCa and PCa was 50 and 43. Is that correct?
We thank the reviewer for the comment.
As noticed, PV values reported in Table 1 for the study population overall were wrong.
The typo was emended as follows:
These correct values, in fact, are in line with previous multicenter cohort studies based on patients from Southern European countries (doi.org/10.1038/s41391-018-0054-9).
- Since the paper correctly mention the imbalance of different hormones (such as oestrogen and testosterone), did the Authors have data concerning the role of levels of these hormones in men with metabolic syndrome and/or prostate cancer? Moreover, since many disorders are associated with metabolic sindrome (doi : 10.2164/jandrol.108.006015), such as hypertension, hyperglycaemia and hypogonadism, and in this latter case testosterone administration has been suggested as potential therapeutic drugs, what is the Authors point of view about the therapeutic use of testosterone? Indeed, in the scientific literature, the testosterone administration and prostate cancer risk is still debated (doi: 10.1158/1055-9965.EPI-04-0715; doi: 10.1016/j.ucl.2007.08.002; doi: 10.1093/aje/kwz138; doi: 10.1177/1756287215597633).
We thank the reviewer for the comment.
Actually, sex hormones levels and possible testosterone replacement therapies were not assessed before prostate biopsy, so that their association with MetS and PCa was not investigated. We acknowledged this limitation and briefly discussed the topic at the end of the Discussion. All the suggested studies were cited.
- (Discussion, pg 7, lines 236-245)
One last limitation of the present study is that sex hormones levels and testosterone replacement therapies (TRT) were not assessed at the time of prostate biopsy: consequently, their association with PCa and MetS was not investigated. However, a strong epidemiologic association between hypogonadism, MetS and its features was recently observed [56]. Besides, there is grounded evidence that sex hormones play a key role in the development and progression of PCa, although the pathogenetic mechanisms was not clarified yet [57]. Also the influence of TRT on tumorigenesis is still debated[58], with recent evidences supporting that it has little, if any, negative impact on the prostate, even in men with a history of PCa [57,59,60].
- Latin words and abbreviations should be written in italics, such as "vs" at line 109, page 3.
We thank the reviewer for the suggestion.
All the abbreviations of “versus” are now displayed as vs.
Also the term “metaflammation” (pg 5, line 161) is now written in italics.
- The p value should be written as < 0.05 or <0.01.
We thank the reviewer for the suggestion.
We went through the entire manuscript and checked for spaces and decimals: all the p values are now displayed properly, both in the text and in the tables.

Round 2
Reviewer 1 Report
I congratulate you for the great effort made to improve the document.
Reviewer 2 Report
Authors have incorporated all suggestions.
Reviewer 3 Report
Dear Authors,
The manuscript has been improved following the reviewers suggestions and should be considered for publication.